# Localization of Receptors for Sex Steroids and Pituitary Hormones in the Female Genital Duct throughout the Reproductive Cycle of a Viviparous Gymnophiona Amphibian, *Typhlonectes compressicauda*

**DOI:** 10.3390/ani11010002

**Published:** 2020-12-22

**Authors:** Claire Brun, Jean-Marie Exbrayat, Michel Raquet

**Affiliations:** 1Sciences and Humanities Confluence Research Center, UCLy, CEDEX 02, 69288 Lyon, France; mraquet@univ-catholyon.fr; 2Ecole Pratique des Hautes Etudes, Paris Sciences Lettres, CEDEX 02, 69288 Lyon, France

**Keywords:** steroid receptors, gonadotropin receptors, prolactin receptors, oviduct, caecilian

## Abstract

**Simple Summary:**

Females of the legless amphibian Cayenne caecilian *Typhloneces compressicauda* demonstrate a biennial viviparous reproductive cycle, with complex morphological alterations in its oviduct. During the first year, these morphological variations permit the capture of the oocytes at ovulation and the pregnancy in the posterior part transformed into uterus. Pregnancy lasts 6 to 7 months and, at parturition, the female gives birth to 6 to 8 newborns which look like small adults.The second year of the cycle is a sexual rest period, allowing females to replenish their body reserves. The hormonal receptors detected in the different cell types of the oviduct confirm that the cyclical development of the genital tract is dependent on sex and pituitary hormones, with a direct control by the pituitary gland.

**Abstract:**

Reproduction in vertebrates is controlled by the hypothalamo-pituitary-gonadal axis, and both the sex steroid and pituitary hormones play a pivotal role in the regulation of the physiology of the oviduct and events occurring within the oviduct. Their hormonal actions are mediated through interaction with specific receptors. Our aim was to locate α and β estrogen receptors, progesterone receptors, gonadotropin and prolactin receptors in the tissues of the oviduct of *Typhlonectes compressicauda* (Amphibia, Gymnophiona), in order to study the correlation between the morphological changes of the genital tract and the ovarian cycle. Immunohistochemical methods were used. We observed that sex steroids and pituitary hormones were involved in the morpho-functional regulation of oviduct, and that their cellular detection was dependent on the period of the reproductive cycle.

## 1. Introduction

Sex steroid hormones exert various physiological effects and, particularly, participate in reproductive functions in all vertebrates [1]. Their action depends on the interactions between the hypothalamus, adenohypophysis and gonads, and involves complex stimulatory and inhibitory pathways. The classic (but not exclusive) mode of action of the steroid hormones is a genomic pathway, involving specific intracellular receptors that are members of the nuclear receptor superfamily. The α and β estrogen receptors (ERα, ERβ) and progesterone receptor (PR) act as ligand-dependent transcription factors and allow to regulate diverse target genes, regulating reproduction [2,3]. The pituitary hormones, the follicle-stimulating hormone (FSH) and luteinizing hormone (LH), are the major components that regulate gonadal production of steroid hormones via their specific receptors, follicle-stimulating hormone receptor (FSHR) and luteinizing hormone receptor (LHR). Both receptors belong to a super-family of receptors that interact with G-proteins, resulting in the stimulation of the membrane-bound adenylyl cyclase that elevates intracellular cyclic adenosine monophosphate (cAMP) concentration [4]. The Prolactin, secreted by the anterior pituitary gland, is also involved in different biological activities, including reproduction. The prolactin receptor (PRLR) is a member of the growth hormone receptor superfamily [5]. These different gonadal and pituitary hormones are involved in the regulation of the morphological changes of the genital tract throughout the reproductive cycle. The study of the expression of their receptors in the oviduct provides information on the role of hormones in the regulation of its gamete and embryos transport, fertilization, and early embryonic development. 

The reproductive cycles of a large number of amphibians are now well known, but hormonal data are available for only a few species of them [6]. Particularly, hormonal influence on the plasticity of the genital tract of Gymnophiona are not well known. Gymnophiona, also called Caecilians, are one of the three orders of Amphibia, whose mode of reproduction is either viviparity or oviparity. Their genital tract therefore reflects this diversity of reproduction patterns [7,8]. *Typhlonectes compressicauda* is one of the 50% of the viviparous species, whose genital tract has been well described [9,10,11,12]. It is divided into two parts after the ostium (funnel), which is an elongated gutter, parallel to the oviduct and ovary. The anterior part of the oviduct is short and flexuous, with glands involved in the synthesis of egg envelopes. The posterior part is longer, differentiated as uterus, in which embryos develop [12,13]. As in many species, the genital tract of *Typhlonectes compressicauda* undergoes significant morphological, and physiological changes during the reproductive cycle of females. This last is biennial, closely linked to the seasonal alternations characterized by a dry season from July till December and a wet season from January till June. The first year of the cycle begins in October, during the dry season, with vitellogenesis. This one continues until January (preparation phase for reproduction). Mating and fertilization take place in February (sometimes until April) and parturition between July and September (reproduction period). After parturition, a new vitellogenesis is observed but ovulation does not occur at the theoretical time. A period of sexual rest (quiescence) is maintained until next October, which marks the beginning of a new cycle. At ovulation, the ostium develops and the gutter inserts the ovary so as to collect oocytes as soon as they are released after rupture of the follicles. At the opposite, it is poorly developed during sexual quiescence and bordered with a layer of undifferentiated cells [10]. Before ovulation, the connective tissue of the anterior part of the oviduct contains numerous cells; it is richly vascularized and sends crests into the lumen; glandular cells develop between the folds of the crests; some ciliated cells are also observed in the epithelium. The uterine part also increases significantly; the wall is bordered with vascularized crests; the epithelium develops and both ciliated cells and two types of glandular cells are observed. At ovulation, cilia of the anterior and posterior parts are covered with a layer of mucus. During pregnancy, the tubal part gradually involutes, while the uterine wall is subjected to a series of variations closely related to the stages of embryo development. In particular, the uterus emits abundant secretions participating in the nutrition of embryos [12,13]. After parturition, the oviduct becomes quiescent, with a narrow lumen bordered with undifferentiated epithelium surrounded with a thin connective wall. At the start of the second year, a new evolution of the genital tract is observed, but it stops at the theoretical period of ovulation, and the oviduct remains quiescent till the next period of preparation of reproduction. These different morphological variations reflect the functionality of genital tract during the sexual cycle, i.e., capture of oocytes at ovulation and embryonic development. These variations correspond to tissue remodeling, implying cell proliferation, apoptosis and cell differentiation. Raquet et al. showed that cell proliferation is substantial during the preparation for reproduction in the oviduct of *Typhlonectes compressicauda*, while apoptosis is mainly involved in the regression of genital ducts after parturition and during the period of sexual rest [11].

The ovarian follicles are equipped to synthesize steroid hormones. Histochemical detection of Δ5 3β hydroxy steroid dehydrogenase, involved in the steroid biosynthesis, has given positive results in both ovarian follicles with vitellogenic oocytes and in corpora lutea [14]. The use of antibodies directed against estriol and 17β-estradiol has shown that labeling is mainly observed in granulosa of follicles containing vitellogenic oocytes [15]. Finally, the use of radioactive pregnenolone has shown that corpora lutea have an endocrine activity linked to pregnancy, like in other viviparous gymnophiona [16,17]. As in other amphibians, reproduction in female is under pituitary control [18,19,20,21]. Variations of gonadotropic (LH and FSH) and lactotropic cells have been observed throughout the sexual cycle of *Typhlonectes compressicauda* [21,22,23,24,25]. All these cells develop from October until February, reach a maximal size in April in pregnant females, when embryos begin to be fed with uterine secretions, and decrease to reach a minimal size at parturition. In non-pregnant females, the size of gonadotropic and lactotropic cells abruptly decreases in February. Moreover, the presence of prolactin-receptors has been detected in ovaries.

These different observations led the authors to propose a scheme of hormonal regulation of female sexual cycle [23]. In ovary, during vitellogenesis, follicle cells became steroidogenic. They are filled with steroid estrogens, which permit regulation of development of the genital tract. The threshold allowing ovulation may also be reached with progesterone. After ovulation, the empty follicles become corpora lutea, the granulosa cells invade the central cavity, cells originating from the theca infiltrate between the granulosa cells, and a neo-vascularization develop. Some cells present the capacity to synthesize progesterone. At the end of pregnancy, the regression of corpora lutea is correlated with the birth of offspring and with the return to a resting state of the genital tract. So, the endocrine activity of corpora lutea seems to be responsible for maintenance of pregnancy. In hypophysis, cellular morphological variations of gonadotropic and lactotropic cells are also correlated to ovary activity. Pituitary hormone cells increase in size and number during vitellogenesis and at the beginning of pregnancy, then decrease notably. During the second year, the pituitary endocrine cells are well developed before the theoretical period of ovulation, then regress quickly, concomitantly with the follicle degeneration. Differentiation of genital tract is prevented [23].

The aim of this study was to specify some aspects of the regulation of female genital tract in this viviparous species of Gymnophiona. For that, we searched to visualize the presence of the steroid hormone receptors (estrogen and progesterone receptors) and pituitary hormone receptors (gonadotropin and prolactin receptors) in the different parts of oviduct throughout the reproductive cycle. The objective being to better understand the importance of hormonal balance in modulation of the morphology of genital tract during the year.

## 2. Materials and Methods 

The animals belong to a collection of *Typhlonectes compressicauda*, captured in the swamps of Kaw in French Guyana throughout the years 1979 to 1987, and stored at the general laboratory of Lyon Catholic University. At this time, no authorization was required to collect animals in the field. The animals were immediately fixed in Bouin’s fluid, and stocked in 70% ethanol, or formalin [26]. Genital tracts were dissected from these fixed animals [10]. Due to the scarcity of biological material, only two females from each period of the sexual cycle were selected, i.e., two pregnant females, two quiescent females and two females ready for breeding. In total, the reproductive systems of six females were studied. The genital tracts were dissected, included in paraffin and cut into 5 µm sections. For the different sections of the genital tract, two slides were used to perform histochemistry. 

Indirect immunohistochemical method has been chosen to localize hormone receptors. Several studies have shown there was an acceptable conservation of the biochemical composition of antibodies in all vertebrates [1,27,28,29,30,31,32,33,34]. According to these studies, we have chosen to use antibodies generally used to detect the hormones in several animal species. Selected sections were deparaffinized, rehydrated in 0.1 M of Phosphate Buffered Saline (PBS) pH 7.4, subjected to antigen retrieval using a microwave 700 w, for 7 min (Antigen unmasking solution, H-3300, Vector laboratories, Les Ullis, France), then incubated in 3% H_2_O_2_ for blocking endogenous peroxidase, and washed in PBS for 5 min. The sections were then incubated for 10 min with bovine serum albumin (BSA), in order to block non-specific binding of antibodies, and stained for 1 h at room temperature with primary antibodies directed against either the progesterone receptor (1:50, AbCys C 10-7068), the ERα and ERβ estrogen receptors (1:50, Santa Cruz sc-7207 and 1:50, Santa Cruz sc-8974, respectively), the gonadotropin receptor (1:50 for FSHR, Santa Cruz sc-13935 and 1:50 for LHR, Santa Cruz sc-25828) or prolactin receptor (1:100, Santa Cruz sc-20992). After washing the slides in PBS, the immunoreactions were visualized with a streptavidin-biotin amplification kit using Amino Ethyl Carbazole (AEC) as chromogen (Kit VECTASTAIN Elite ABC kit, Vector laboratories, Peterborough, UK). The slides were counterstained with hematoxylin QS, a modified Mayer’s hematoxylin, and the preparations were mounted with VectaMount AQ Mounting Medium. Some negative controls (deletion of primary or secondary antibodies) were performed in order to determine specific and non-specific reactions. Histological preparations were photographed using a Nikon Eclipse E400 light microscope, equipped with a Nikon digital camera DXM1200, connected to a computer with Nis-Element BR3.1 software (Lucia software). Hormone receptor results are expressed as a percentage of labeled cells. 

For statistical analysis, two slides of each genital tract were observed, and the number of labeled cells per thirty cells was determined for each cell type. A quantitative study was made by using the statistical option of the XLSTAT Excel data analysis (Microsoft Excel extension). The number of cells (*n* = 100) measured for each category and for each antibody was always higher than the number obtained by the formula (standard deviation·0.83/mean·0.05)^2^, except for a few cases where no significative difference was registered. All data were analyzed by analysis of variance (ANOVA) and a Fisher test (LSD) to determine differences among groups (*p* ≤ 0.05).

## 3. Results

## 3.1. Detection of Sex Steroid Receptors

### 3.1.1. Progesterone Receptors (PR)

In ostium

The immunohistochemical profile of progesterone receptors showed a low percentage of nuclei labeled in the ostium. Before ovulation, the percentage of labeled cells was 0% in connective cells and 17.8% in ciliated cells. During the reproduction period, it was 0% and 7.6% in connective cells for pregnant and quiescent females, respectively, and 4.5% and 14.5% in ciliated cells for pregnant and quiescent females, respectively. The number of nuclear PR detected in the different cell types was significant through the sexual cycle (*n* = 6, Pr > F: 0.028 for ciliated cells and Pr > F: 0.003 for connective cells) (Figure 1A).

In tubal part

The percentages of labeled cell nuclei varied according to the tissue’s structures and period of the reproductive cycle. In connective cells, the percentage of PR increased significantly between the period of preparation for breeding and the reproduction period (*n* = 4, Pr > F: 0.039), the difference being not very significant between pregnant and quiescent females (*n* = 4, Pr > F: 0.044). In secretory cells, the proportion of labeled cells were 26.2% before ovulation, 17.5% and 11.8% in pregnant and quiescent females, respectively. These variations were significant between the period of preparation for breeding and pregnant period (*n* = 4, Pr > F: 0.036), but not between the pregnant and quiescent females (*n* = 4, Pr > F: 0.246). Regarding the ciliated cells, a maximal percentage of cell nuclei was labeled during the preparation of reproduction (48%); it decreased during the pregnancy (26.8%) and even more during the sexual rest period (11%). These variations in the ciliated cells were significant (*n* = 6, Pr > F: 0.007) (Figure 1B and Figure 2A).

In uterus

In the fibroblasts, nuclear PRs were detected before ovulation and in pregnant females, with a similar percentage (about 15%) (*n* = 4, Pr > F: 0.622). No receptor was detected in quiescent females. Thus, contrary to what was observed during the first year, the evolution of nuclear PRs was significant for the fibroblasts during the second year of the sexual cycle (*n* = 4, Pr > F: 0.012). In secretory cells, the proportion of labeled nuclei was notable during preparation for reproduction (27.5%), breeding period (33.8%) and sexual rest period (21.6%). In ciliated cells, an important percentage of nuclear PR was detected during the preparation period (47%). During pregnancy, the number of such receptors decreased abruptly to reach 0%, while it stood at high level during the sexual rest period (23.5%). All these variations observed were statistically significant (*n* = 6, Pr > F: 0.042, Pr > F: 0.001 for secretory cells and ciliated cells, respectively) (Figure 1C).

## 3.1.2. Estrogen Receptors (ERα and ERβ)

The nuclear ER were widely detected in the various cell types of the genital duct, however notable differences in their expression patterns were observed during the sexual cycle.

In ostium

The percentage of nuclear ER detected in connective cells decreased abruptly after preparation for reproduction period and came down to 0% in pregnant and quiescent females (*n* = 6, ERα: Pr > F: 0.000; ERβ: Pr > F < 0.0001). Similarly, the number of ER detected in ciliated cells was significant before ovulation (ERα: 19.2%, ERβ: 35%) and only ERβ was detected during pregnancy (38.3%), while only ERα alone was detected in sexual rest (26.2%). These differences, observed in the ciliated cells, were significant (*n* = 6, ERα: Pr > F: 0.07; ERβ: Pr > F: 0.000) (Figure 3A and Figure 4A).

In tubal part

When the oviduct was prepared for breeding, nuclear ERα was detected in 32% of connective cells, 47.8% of secretory cells, but not in ciliated cells. During the breeding period, these percentages decreased in the different cell types, except in ciliated cells whose number of nuclear ERα increased. In quiescent females, no nuclear ERα was detected in any tissue. These percentage differences observed through the sexual cycle were significant (*n* = 6, Pr > F: 0.001, Pr > F: 0.000, Pr > F: 0.005 for connective cells, secretory cells and ciliated cells, respectively). 

The evolution of nuclear ERβ in the different tissue of tubal part during the reproductive cycle was different from that of nuclear ERα. The percentage of anti-nuclear ERβ positive connective cells changed significantly through the cycle; it rose from 10% before ovulation to 31.5% during pregnancy, and fell down to 0% during the sexual inactivity period in quiescent females. In secretory cells, 31% of cells were immunostained before ovulation, 25% during pregnancy and 0% during the theorical period of reproduction in the quiescent females. Staining in ciliated cells was exclusively observed when the oviduct was prepared for breeding. The evolution of the number of ERβ through the sexual cycle was significant (*n* = 6, Pr > F: 0.004, Pr > F: 0.001, Pr > F: 0.000 for connective cells, secretory cells, and ciliated cells, respectively) (Figure 3B and Figure 4B).

In uterus

As for the other parts of the oviduct, immunohistochemical profile of uterine tissue changed depending on the seasons. The two nuclear ER isoforms were detected in the different cell types during the period of preparation for reproduction. In pregnant females, it should be noted that no ERα was detected in secretory cells, when 75% of these cells were ERβ-positive. No ERβ was expressed in ciliated cells, contrary to the ERα (33%). Results were different in quiescent females, in which only nuclear ERα was detected both in connective cells (33% of cells) and secretory cells (37.5% of cells). The evolution of the number of ERs through the sexual cycle was significant (*n* = 6, ERα: Pr > F: 0.001, Pr > F: 0.001, Pr > F: 0.001 for connective cells, secretory cells and ciliated cells, respectively; ERβ: Pr > F: 0.003, Pr > F: 0.001, Pr > F: < 0.0001 for connective cells, secretory cells and ciliated cells, respectively) (Figure 2B,C, Figure 3C and Figure 4C).

## 3.2. Detection of Gonadotropin Receptors (FSHR and LHR) and Prolactin Receptor (PRLR)

In ostium

Before ovulation, 22.6% of connective cells and 36.2% of ciliated cells of the oviduct showed anti-FSHR labelling. During pregnancy and sexual rest, the percentage of connective cells decreased to 0%; the percentage of ciliated cells increased to 42.5% in pregnant females but decreased to 25.8% in quiescent females (*n* = 6, Pr > F: 0.002, Pr > F: 0.055 for ciliated cells and connective cells, respectively).

Regarding the detection of LHR, 7.7% of ciliated cells reacted with anti-LHR serum before ovulation. This percentage increased to 51.5% in pregnant females and to 18% in quiescent females during the inactivity period. These differences observed were significant (*n* = 6, Pr > F: 0.002). A weak signal for LHR (11%) was detected in the connective cells of quiescent females, unlike what was observed during other periods of the sexual cycle (*n* = 6, Pr > F: 0.001).

Prolactin receptors were detected in the different cell types according to the period of sexual cycle. In connective cells, 18% of cells showed positive immunoreactivity during the preparation period, 0% during reproduction in pregnant females and 23.6% in quiescent females (*n* = 6, Pr > F: 0.002). In ciliated cells, the percentages were, respectively, 13.5%, 8.3%, and 15.5% before ovulation, in pregnant, and quiescent females; these percentages did not statistically vary before and after ovulation (*n* = 6, Pr > F: 0.158) (Figure 5A, Figure 6A and Figure 7A).

In tubal part

Before ovulation, abundant membrane cells showed positive signal for FSHR in the different cells: 34.1% in connective cells, 32.5% in secretory cells, 42.7% in ciliated cells. This immunohistochemical profile remained constant during the breeding period (pregnant females) and was not statistically significant except for the connective cells (*n* = 4, Pr > F: 0.003; Pr > F: 0.071, Pr > F: 0.835 for secretory cells and ciliated cells, respectively). On the other hand, the percentage of labeled cells for FSHR decreased in quiescent females (*n* = 4, Pr > F: 0.009, Pr > F: 0.035, Pr > F: 0.001 for connective cells, secretory cells, and ciliated cells, respectively).

The LH receptors were weakly detected during the preparation for reproduction in both connective and secretory cells (5.8% and 5.9%, respectively) and not at all in ciliated cells. During the breeding period (pregnant females), the percentage of labeled cells for LHR remained stable in secretory cells (6.5%), increased significantly in ciliated cells (20.3%) and fell down to zero in connective cells. No LHR was detected in the oviduct of quiescent females. These differences were significant (*n* = 6, Pr > F: 0.005, Pr > F: 0.031, Pr > F: 0.001 for the connective cells, secretory cells, and ciliated cells, respectively).

Percentage of PRLR-positive connective cells decreased from preparation period (22.5%) to reproductive period in pregnant females (11.5%), but not in quiescent females (32.5%) (*n* = 6, Pr > F: 0.013). In secretory cells, 41.6% of cells were stained when the oviduct was prepared for breeding, 8.5% in pregnant females but 27.5% in quiescent females (*n* = 6, Pr > F: 0.000). In ciliated cells, the number of PRLR detected fluctuated slightly during the sexual cycle and the differences between the values were not significant (*n* = 6, Pr > F: 0.095) (Figure 5B, Figure 6B and Figure 7B).

In uterus

The percentage of FSHR labeled cells varied slightly in connective cells during the reproductive cycle, and this evolution was not significant (*n* = 6, Pr > F: 0.575). The percentage of FSHR-labeled secretory cells increased significantly during the period of reproduction; it evolved from 13.5% during preparation period to 26% in pregnant females, and 38.3% in quiescent females (*n* = 6, Pr > F: 0.016). The percentage of FSHR-labeled ciliated cells also increased strongly from preparation (19%) to pregnancy (50.3%); it decreased to reach 0% in quiescent females (*n* = 6, Pr > F: 0.002).

The LHR detection evolved differently during the reproductive cycle depending on the cell type. Before the ovulation, the LHR was detected in 20.2% of connective cells, in 13.7% of secretory cells, and never in ciliated cells. During the breeding period, LHR was spotted exclusively in secretory cells, with 42.5% of labeled cells in pregnant females (0% in quiescent females). These differences were significant (*n* = 6, Pr > F: 0.002, Pr > F: 0.001 for the connective cells and secretory cells, respectively).

The percentage of PRLR-positive cells fluctuated significantly during the sexual cycle. In connective cells, it increased from 11% before the ovulation period to 37.5% in the pregnant females or 31.6% in quiescent females. In secretory cells, it rose from 90% in preparation period to about 23% in reproductive period (pregnant females: 26%, quiescent females: 21.5%). In ciliated cells, PRLR was detected only during the reproductive period (pregnancy and quiescent females). All these differences were significant, except for ciliated cells (*n* = 6, Pr > F: 0.165; Pr > F: 0.004 for the connective cells, and Pr > F: 0.007 for the secretory cells) (Figure 2D–F, Figure 5C, Figure 6C and Figure 7C).

## 4. Discussion

Immunohistochemical detection of sex steroid and pituitary hormone receptors was performed in different parts of the genital tract throughout the sexual cycle of *Typhlonectes compressicauda*. To carry out the experiments, antibodies directed against human hormone receptors were used. Beforehand, we made sure that a high sequence homology existed between amphibian and human proteins. In fact, the DNA-binding domain and the ligand-binding domain of the amphibian ERα and ERβ are highly conserved, and very similar to that of other vertebrates [29,30,31]. For example, the level of the amino acid sequence homology in the putative DNA binding region in the human and *Xenopus laevis* ERα is greater than 98%, suggesting intense evolution pressure of estrogen response [31]. On the other hand, homology between PRs from different vertebrate species revealed that the central ligand binding domain and the central DNA binding domain shared high amino acid identity (amino acid identity better than 92 and 80%, respectively) [32]. FSHR and LHR in mammals and amphibians were also highly conserved during evolution of vertebrates [33]. Finally, a mouse anti-PRLR antibodies was used to study the frog *Rana ridibunda* protein [34]. These different data allowed us to consider that human anti-steroid or anti-pituitary receptor antibodies can be used for immunolocalization of amphibian hormone receptors.

The results showed that each hormonal receptor was distributed over the oviduct according to the cell type, the part of the oviduct, and period of sexual cycle.

Immunoreactive PRs have been identified in the oviductal tract before ovulation and during the breeding season in pregnant and quiescent females. The specific profile observed during the sexual cycle for each type of cell appeared to be consistent with the functions performed by the oviduct, particularly concerning the known role of progesterone during ovulation and during the preparation of the uterus for gestation. Before ovulation, a high percentage of epithelial cells were labeled with antibodies directed against progesterone receptors, especially in the tubal part and uterus. These results may be explained by a proliferative role of progesterone. A previous study showed that before ovulation, the rate of epithelial cells proliferation was higher than the rate of apoptosis in different parts of the oviduct [11]. During gestation, the percentage of anti-PR secretory cells decreased in tubal part, whereas it increased in uterus. The higher sensitivity to progesterone during gestation of these epithelial cells argues for an active role of this hormone in the maintenance of gestation and in the development of uterine secretions which participate in the nutrition of the embryos.

The presence of nuclear estrogen receptors has been investigated in cells of the genital tract. Differential expression of ERα and ERβ was observed depending on cell type and sexual cycle period. The percentage of anti-ERα labeled cells was low when the percentage of anti-ERβ labeled cells was high, and reciprocally. Particularly, the balance of ERα and ERβ was in favor of ERβ in the secretory cells, and in favor of ERα in the ciliated cells in the tubal part and the uterus of pregnant females. In quiescent females, ERβ was not detected in any cells of the oviduct and ERα was not detected in the cells of the tubal part. 

These results confirm that the two ERs have a functional role during the sexual cycle, as in other vertebrates [35]. Whereas ERα is exclusively detected in ciliated epithelial cells, ERβ is expressed in secretory cells of pregnant females. This specific distribution would illustrate the specific roles of these two ERs during reproduction. In accordance with this hypothesis, experiments on mice have demonstrated the action of ERα in the oviductal transport of embryos [36,37]. 

Reproduction of female Gymnophiona is also under hypophysis control. Gonadotropic cells present morphological variations which correlate to the sexual cycle [19,20,21,22,23]. In *Typhlonectes compressicauda*, the cells with gonadotropic hormone receptors have been detected in the different tissues of the genital tract. In the tubal and uterine parts, the percentage of cell reacting with anti-LHR and anti-FSHR antibodies was variable according to the period of the sexual cycle. It was substantial before ovulation and during the pregnancy, and low or equal to zero during the inactive sexual rest. These results show that the pituitary gland exercises a direct control over the genital tract and that this control is regulated during the sexual cycle. The presence of LHR and FSHR together in the tissues suggests that the two hormones have synergistic action on the development of histological structures and on the stimulation of oviductal secretions during periods favorable to reproduction. Previous studies into the pituitary gland of *Typhlonectes compressicauda* showed that the secretion of gonadotropic hormones was correlated to ovarian activity and to variations of genital tract [21]. Between October and January, the gonadotropic cells were well developed, in correlation with the vitellogenic activity and the development of the oviduct. At the beginning of gestation, gonadotropic cells are still well developed, but progressively are decreased in size and number at the end of pregnancy. During the sexual inactive period, the gonadotropic cells were not developed, in correlation with the degeneration of follicles and regression of tubal and uterine walls.

The role of prolactin in the regulation of the female sexual cycle was also important. Previous immunohistochemistry studies in *Typhlonectes compressicauda* have shown that the size and the number of prolactin cells increase during vitellogenesis and at the beginning of pregnancy. At the end of pregnancy, the volume of these cells gradually decreases and remains minimal during the year of sexual inactivity [21,22]. The results of our experiments showed that the cells of the genital tract react with anti-PRLR serum during the different periods of the sexual cycle. As in the newt *Cynops pyrrhogaster*, PRL seems to be involved in oviductal development and to be essential for the oviductal jelly secretion [19,27,38]. The combination of prolactin and estrogen seems to be necessary. During the breeding season, the connective cells reacting with anti-PRLR serum were the most numerous in the uterus. During the period of sexual rest, hormonal desensitization of the genital tract should go through a loss of PRLR, but their number in connective cells remained high. Similarly, the percentage of positive epithelial cells fluctuated slightly between pregnant and quiescent females. A previous study has indicated that mRNA coding for PRLR were strongly expressed in corpora lutea in the middle of gestation, while during the other periods of the cycle, the expression of these mRNAs was equal and lower [22,23]. With regard to the genital tract, it seems that the morphological changes preceding gestation are under PRL control at ovulation and during the beginning of breeding.

## 5. Conclusions

This immunohistochemical study demonstrated that sexual and pituitary hormone receptors are present in the cells of the female genital tract of *Typhlonectes compressicauda*. During the sexual cycle, each part of the oviduct and each type of cell has a specific sensitivity to hormonal controls. This sensitivity appears to correlate with the preparation of the tubular section for ovulation and the uterus for pregnancy. Before ovulation, the follicles secrete estrogen and progesterone, hormones involved in triggering ovulation and in regulating the development of the genital tract. Progesterone receptors and estrogen receptors are detected in the different parts of the oviduct. During the reproductive period, in pregnant females, the corpora lutea synthesize progesterone, which allows the maintenance of uterus. Steroid hormone receptors are weakly immunodetected in the ostium, which confirms that their role is less significant after ovulation in this first part of the oviduct. In addition to the expected PRs, many ER have been detected in both tubal and uterine parts, which confirms that they are required for gestation, in particular for the oviductal transport of embryos.

The presence of pituitary gonadotropin receptors in the oviduct and changes in the amount of receptor depending on the hormonal status of the females show that these hormones directly regulate tubal function. The detection of PRLRs in the oviductal cells confirms the important role of prolactin in regulation of the female sexual cycle.

## Figures and Tables

**Figure 1 animals-11-00002-f001:**
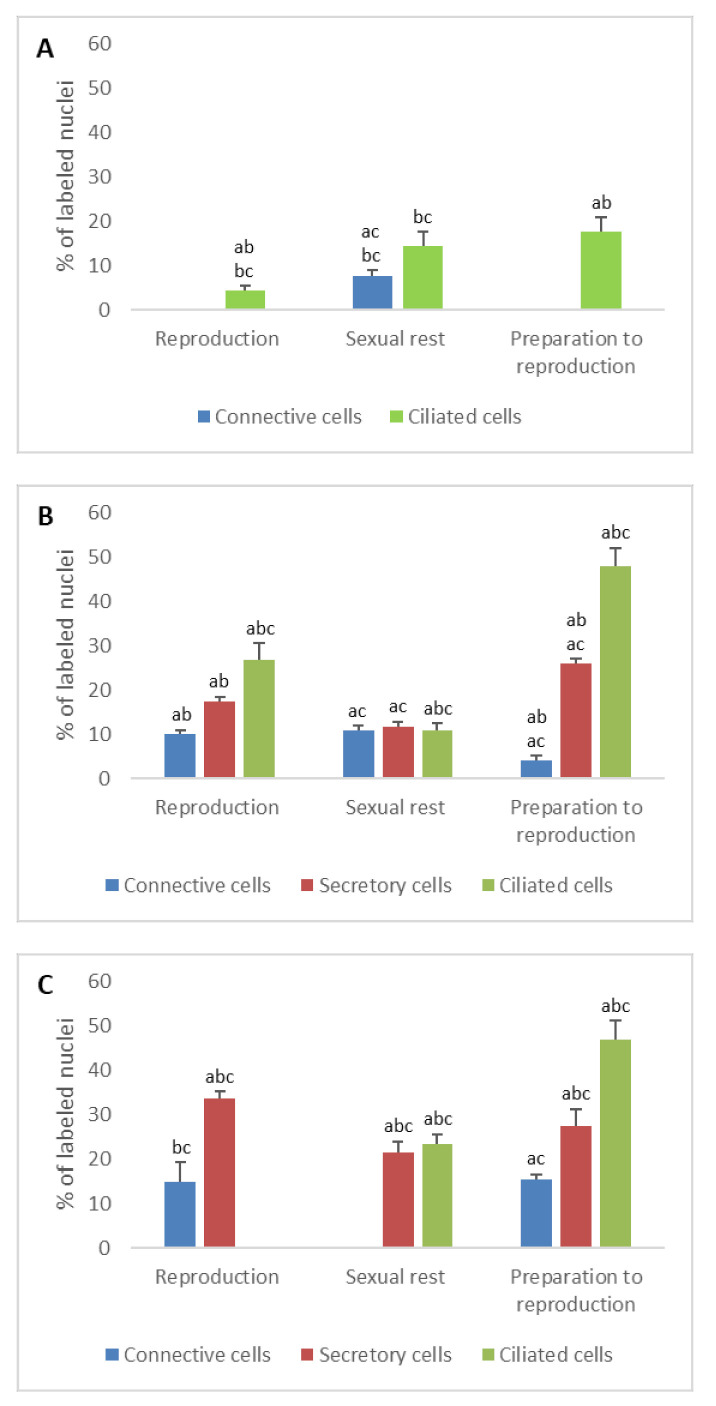
Percentage of PR immunodetected in the different cell types of the genital tract of *Typhlonectes compressicauda* before ovulation and during the reproduction period (pregnant females) and sexual rest (quiescent females). Significant differences (ANOVA, α = 0.05) between the 3 groups of females (a: before ovulation; b: pregnant females; c: quiescent females) are reported with the letters a, b, c. (**A**) Ostium (connective cells *n* = 6, Pr > F: 0.003, ciliated cells *n* = 6, Pr > F: 0.028); (**B**) Tubal part (connective cells *n* = 6, Pr > F: 0.072, secretory cells *n* = 6, Pr > F: 0.078, ciliated cells *n* = 6, Pr > F: 0.007); (**C**) Uterus (connective cells *n* = 6, Pr > F: 0.021, secretory cells *n* = 6, Pr > F: 0.042, ciliated cells *n* = 6, Pr > F:0.001).

**Figure 2 animals-11-00002-f002:**
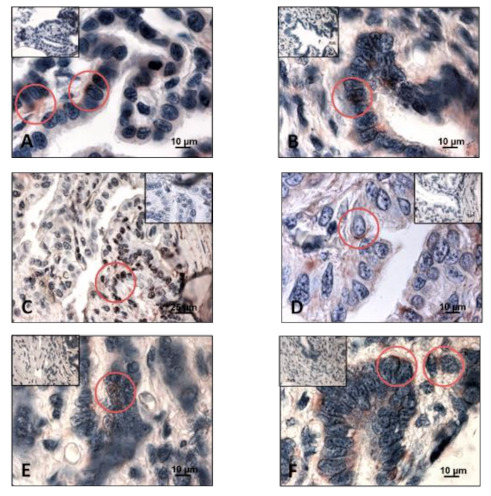
Examples of immunohistochemical reactions in *Typhlonectes compressicauda* genital tract. Red circles show positive labeling. Inserts show negative control. (**A**) Immunodetection of nuclear PR in cells of tubal part during the reproduction period; (**B**) Immunodetection of nuclear ERα in cells of uterus during the preparation of reproduction; (**C**) Immunodetection of nuclear ERβ in cells of uterus during the reproduction period; (**D**) Immunodetection of FSHR in cells of uterus during the reproduction period; (**E**) Immunodetection of LHR in cells of uterus during the preparation of reproduction; (**F**) Immunodetection of PRLR in cells of uterus during the preparation of reproduction.

**Figure 3 animals-11-00002-f003:**
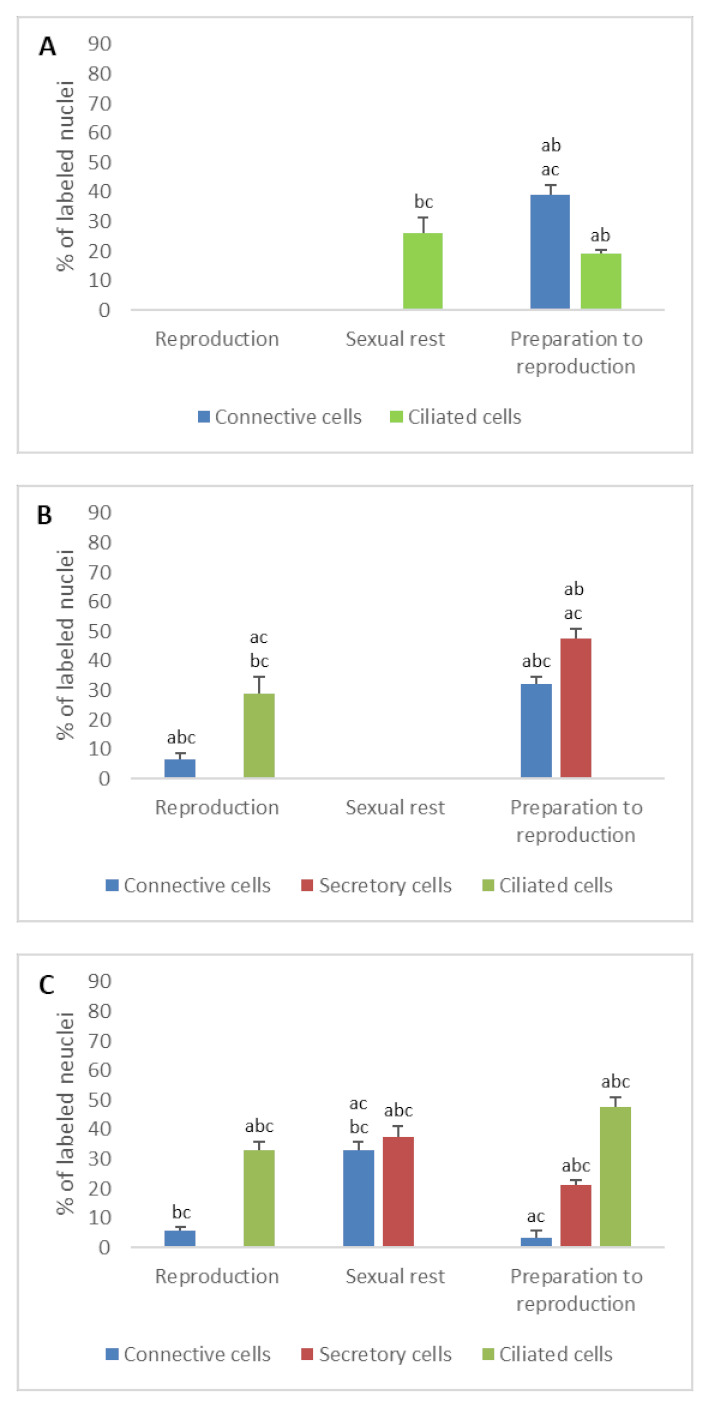
Percentage of ERα immunodetected in the different cell types of the genital tract of *Typhlonectes compressicauda* before ovulation and during the reproduction period (pregnant females) and sexual rest (quiescent females). Significant differences (ANOVA, α = 0.05) between the 3 groups of females (a: before ovulation; b: pregnant females; c: quiescent females) are reported with the letters a, b, c. (**A**) Ostium (connective cells *n* = 6, Pr > F: 0.000, ciliated cells *n* = 6, Pr > F: 0.07); (**B**) Tubal part (connective cells *n* = 6, Pr > F: 0.001, secretory cells *n* = 6, Pr > F: 0.000, ciliated cells *n* = 6, Pr > F: 0.005); (**C**) Uterus (connective cells *n* = 6, Pr > F: 0.001, secretory cells *n* = 6, Pr > F: 0.001, ciliated cells *n* = 6, Pr > F: 0.001).

**Figure 4 animals-11-00002-f004:**
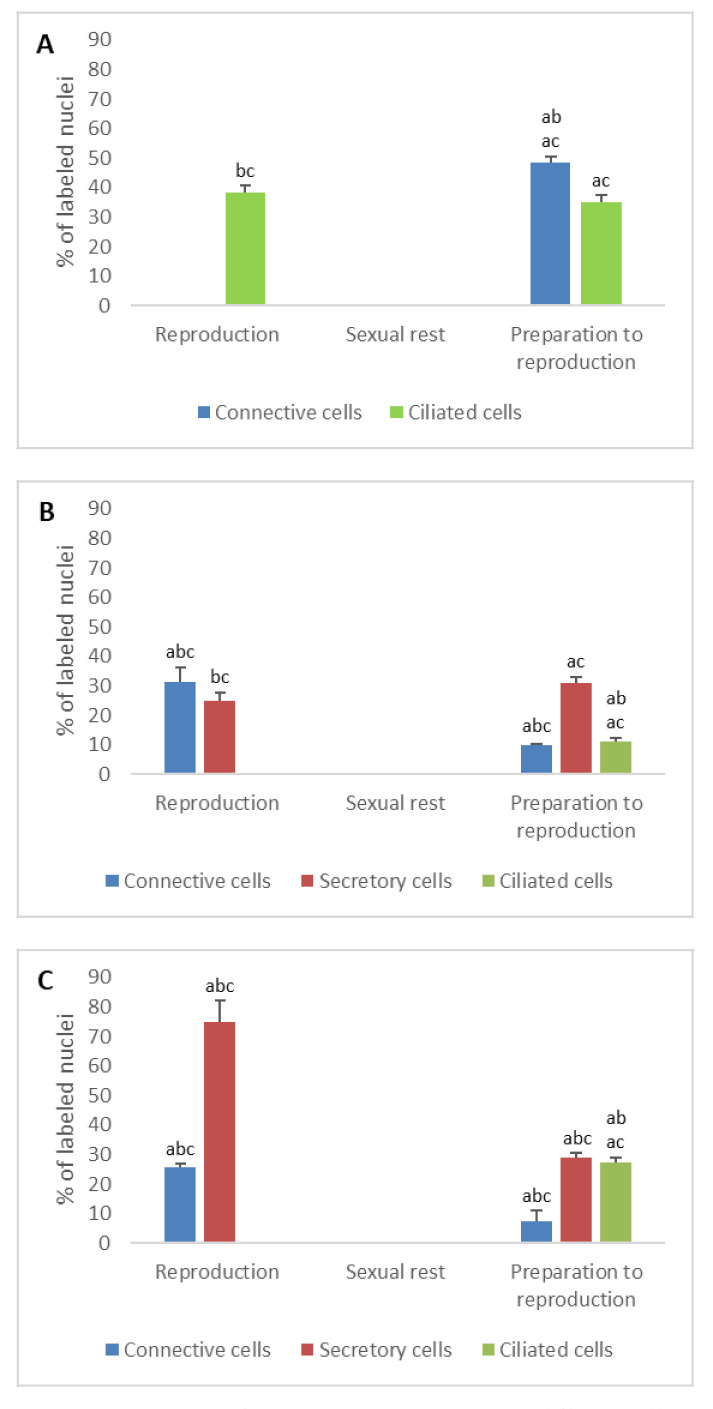
Percentage of ERβ immunodetected in the different cell types of the genital tract of *Typhlonectes compressicauda* before ovulation and during the reproduction period (pregnant females) and sexual rest (quiescent females). Significant differences (ANOVA, α = 0.05) between the 3 groups of females (a: before ovulation; b: pregnant females; c: quiescent females) are reported with the letters a, b, c. (**A**) Ostium (connective cells *n* = 6, Pr > F < 0.0001, ciliated cells *n* = 6, Pr >F: 0.000); (**B**) Tubal part (connective cells *n* = 6, Pr > F: 0.004, secretory cells *n* = 6, Pr > F: 0.001, ciliated cells *n* = 6, Pr > F: 0.000); (**C**) Uterus (connective cells *n* = 6, Pr > F: 0.003, secretory cells *n* = 6, Pr > F: 0.001, ciliated cells *n* = 6, Pr > F: 0.0001).

**Figure 5 animals-11-00002-f005:**
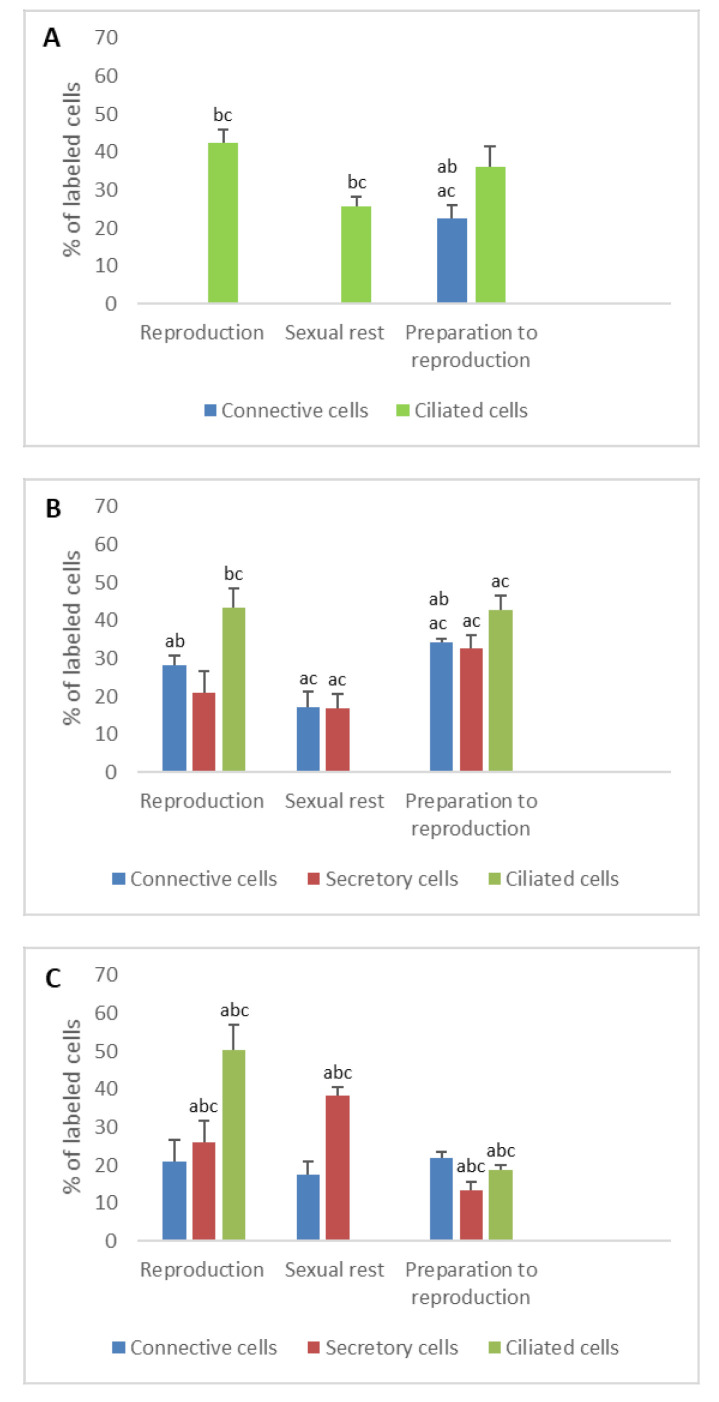
Percentage of FSHR immunodetected in the different cell types of the genital tract of *Typhlonectes compressicauda* before ovulation and during the reproduction period (pregnant females) and sexual rest (quiescent females). Significant differences (ANOVA, α = 0.05) between the 3 groups of females (a: before ovulation; b: pregnant females; c: quiescent females) are reported with the letters a, b, c. (**A**) Ostium (connective cells *n* = 6, Pr > F: 0.002, ciliated cells *n* = 6, Pr > F: 0.055); (**B**) Tubal part (connective cells *n* = 6, Pr >F: 0.006, secretory cells *n* = 6, Pr > F: 0.07, ciliated cells *n* = 6, Pr > F: 0.002); (**C**) Uterus (connective cells *n* = 6, Pr > F: 0.575, secretory cells *n* = 6, Pr > F: 0.016, ciliated cells *n* = 6, Pr > F: 0.002).

**Figure 6 animals-11-00002-f006:**
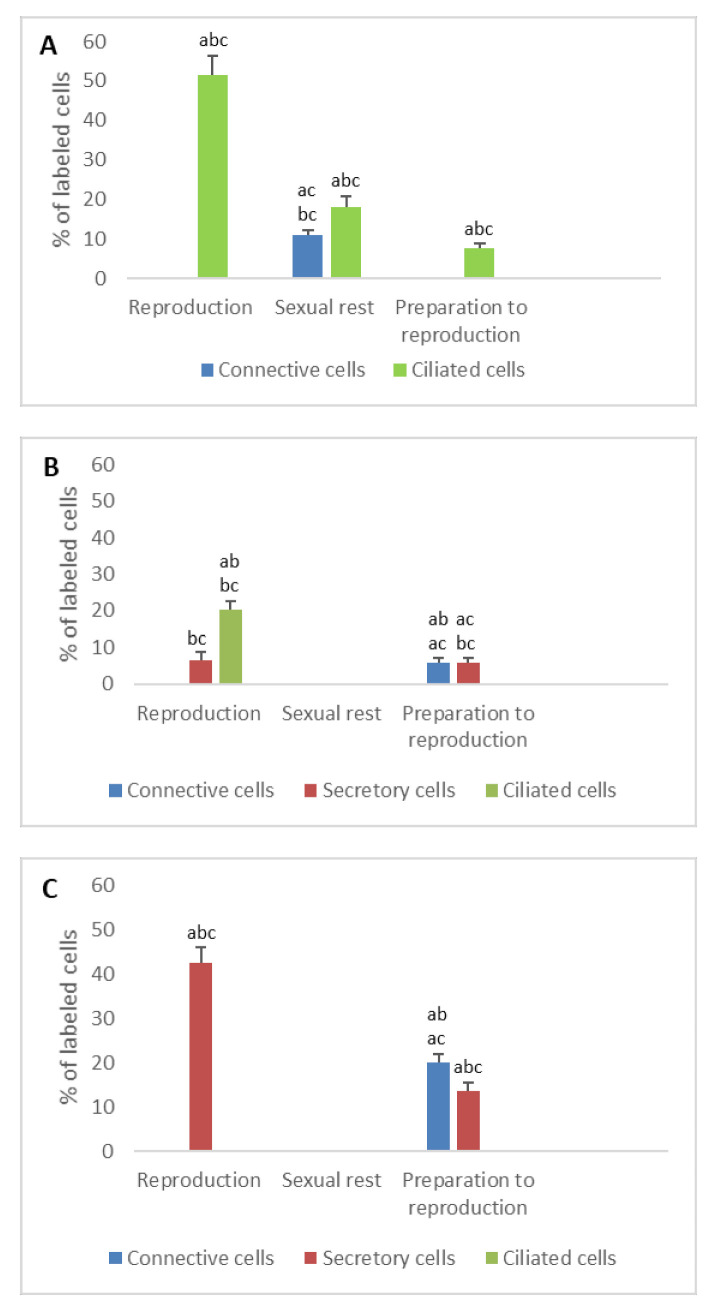
Percentage of LHR immunodetected in the different cell types of the genital tract of *Typhlonectes compressicauda* before ovulation and during the reproduction period (pregnant females) and sexual rest (quiescent females). Significant differences (ANOVA, α = 0.05) between the 3 groups of females (a: before ovulation; b: pregnant females; c: quiescent females) are reported with the letters a, b, c. (**A**) Ostium (connective cells *n* = 6, Pr > F: 0.001, ciliated cells *n* = 6, Pr > F: 0.002); (**B**) Tubal part (connective cells *n* = 6, Pr > F: 0.005, secretory cells *n* = 6, Pr > F: 0.031, ciliated cells *n* = 6, Pr > F: 0.001); (**C**) Uterus (connective cells *n* = 6, Pr > F: 0.002, secretory cells *n* = 6, Pr > F: 0.001, ciliated cells *n* = 6).

**Figure 7 animals-11-00002-f007:**
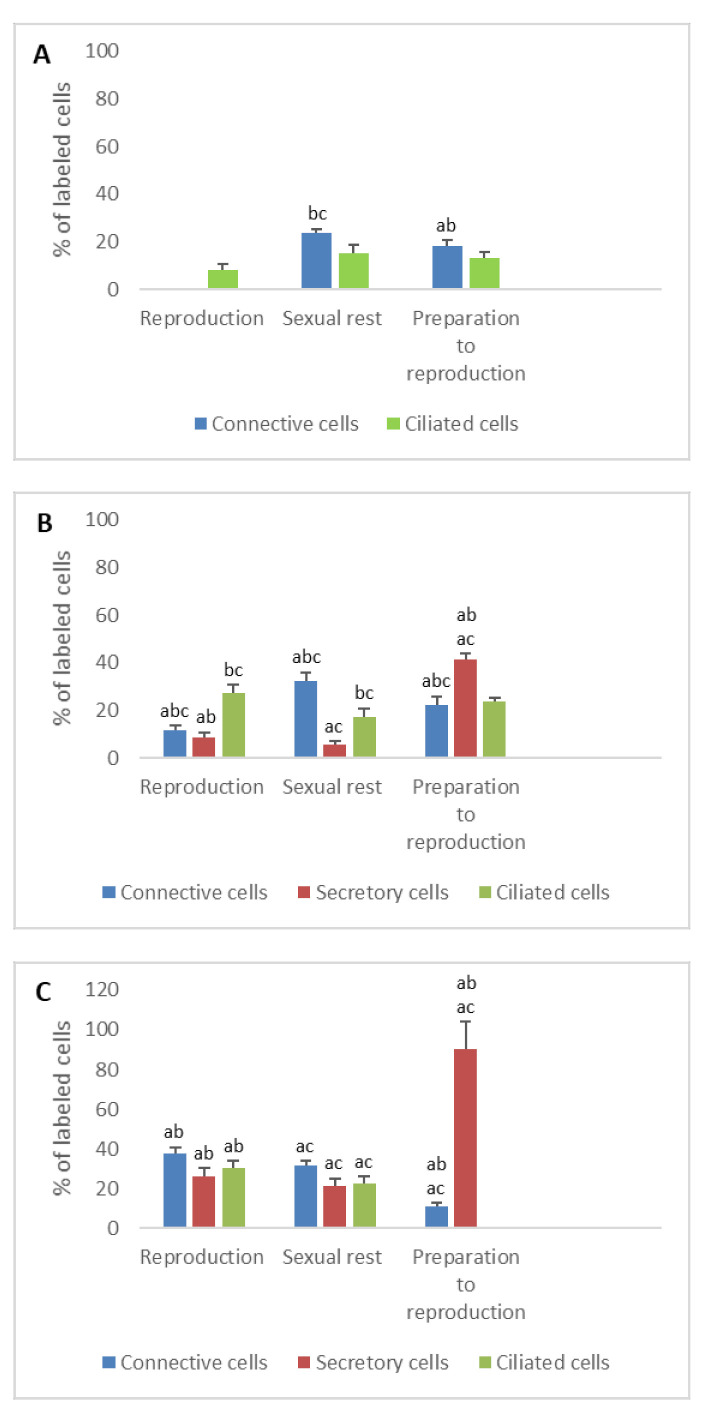
Percentage of PRLR immunodetected in the different cell types of the genital tract of *Typhlonectes compressicauda* before ovulation and during the reproduction period (pregnant females) and sexual rest (quiescent females). Significant differences (ANOVA, α = 0.05) between the 3 groups of females (a: before ovulation; b: pregnant females; c: quiescent females) are reported with the letters a, b, c. (**A**) Ostium (connective cells *n* = 6, Pr > F: 0.002, ciliated cells *n* = 6, Pr > F: 0.158); (**B**) Tubal part (connective cells *n* = 6, Pr > F: 0.013, secretory cells *n* = 6, Pr > F: 0.000, ciliated cells *n* = 6, Pr > F: 0.095); (**C**) Uterus (connective cells *n* = 6, Pr > F: 0.004, secretory cells *n* = 6, Pr > F: 0.007, ciliated cells *n* = 6, Pr > F: 0.165).

## Data Availability

The data presented in this study are available on request from the corre-sponding author. The data are not publicly available because they belong to a research unit with-out a public download system.

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
