# Peer review of "Localization of Receptors for Sex Steroids and Pituitary Hormones in the Female Genital Duct throughout the Reproductive Cycle of a Viviparous Gymnophiona Amphibian, Typhlonectes compressicauda"

_animals, 2020, doi:10.3390/ani11010002_

Round 1

Reviewer 1 Report

Authors have addressed all suggestions. 

Author Response

Dear reviewer,

Thank you for your help for the improvement of the manuscript. Thank you also for your last comment.

Sincerely yours,

Brun, M. Raquet, J.-M. Exbrayat

Reviewer 2 Report

As authors have performed a ANOVA different letters determining differences between the different groups shoudl be included in the graph of data, to clarify were are the statistical differences between groups.

Authors should check some sentence like the ones in lines 140 and 164 as some misspelling have been detected. 

Author Response

Dear reviewer,

Thank you for your help for the improvement of the manuscript. We have corrected the manuscript according to your last remarks.

Comment: “As authors have performed a ANOVA different letters determining differences between the different groups should be included in the graph of data, to clarify were are the statistical differences between groups.”

We have added letters on graphs in order to determine differences between the different groups, with an explanation in the legend. The corrections have been written in red.

Comment: “Authors should check some sentence like the ones in lines 140 and 164 as some misspelling have been detected.”

We have checked the sentences between lines 150 and 164, and corrected the misspellings. The corrections have been written in red.

Sincerely yours,

Brun, M. Raquet, J.-M. Exbrayat

This manuscript is a resubmission of an earlier submission. The following is a list of the peer review reports and author responses from that submission.

Round 1

Reviewer 1 Report

The manuscript describes the hormonal regulation of female genital ducts in a viviparous Gymnophiona Amphibian throughout the reproductive cycle.

By immunohistochemical methods it has been detected: the localization of α and β estrogen, progesterone, gonadotropin and prolactin receptors, in the different parts of the genital tract during the sexual cycle.

Even if the topic of the manuscript is interesting and original, the presentation of immunohistochemical reactions are scarce and also the images are of poor quality (fig 7). The authors should prepare a more clear and convincing figure(s) showing the immunopositivity represented in other figures (histograms).

Further, according to my opinion a broader bibliographic framework including some references of other species in relation to this topic should be mentioned.

Reviewer 2 Report

The manuscript entitled "Some aspect of hormonal regulation of female genital duct throughout the reproductive cycle in Typhlonectes compressicauda (Dumeril and Bibron, 1841), a viviparous Gymnophiona Amphibian"The titled manuscript describes the expression of different hormonal receptors throughout the genital tract of Typhlonectes compressicauda. 

The work is interesting, although the authors must take into account various aspects:
- Firstly, I suggest that the title should be changed since the authors observe the expression of hormonal receptors but not the hormonal regulation. The authors do not determine steroid hormone concentrations.
- The aim of the study must be included in the abstract.
- The authors describe that the ovary is the only organ that produces steroid hormones. Is there no other organ in the genital tract capable of producing steroid hormones, i.e. uterus?
- In the introduction section, the authors should expose the different parts in which the genius female tract of Typhlonectes compressicauda. .
- The aim of the study is not clear and is difficult to understand, I suggest that it should be more specific.
- Regarding the material and methods, the authors should specify the total number of animals they have used. In addition, they should specify if the animals were fixed in the years they cited or were fixed later.
- In the part of immunohistochemistry, how many fields have been counted to do the statistics? How was the staining assessment performed? How was the use of these anti-bodies validated for amphibian samples? Did they use a validated positive control apart from a negative control?
- In the figure legends the authors state that gaphic A corresponds to the oviductal tunnel. However, in the text the authors mentioned ostium. It should be unified to facilitate compression.
-As for all graphs, no reference to statistics is made. I suggest that the authors mark the statistical differences on the graphs, i.e., by adding asterisks.
- In figure 7 the immunoexpression is not well appreciated in some images. Could you increase the resolution of the figure?
- The discussion is too short compared to the introduction. Trying to correlate your results with the hormonal data provided by other researchers can give strength to the discussion and add relevance to your results.
- Could the bibliography used be a little more current?

Reviewer 3 Report

The main problem of this article is that authors did not demonstrate that the antibodies are really detecting what they have to detect as they are not specific antibodies against Typhlonectes compressicauda receptors. In that sense, authors should include the high homology sequence of Typhlonectes compressicauda receptor and the protein used to produce the antibody. It was impossible to find the antibodies reported by the authors in the web of Santa Cruz enterprise, so it is not known the specificity of the antibodies used in this paper.
It is highly important to know if the antibodies are detecting the right proteins, as not all antibodies cross-react between different species.
Moreover, in addition to include the homology of the different proteins in the paper, the authors should also performed a blocking assay with the specific protein or the antigenic peptide or with a high expressed tissue homogenize of Typhlonectes compressicauda or so.
Moreover, authors have to show a representative section of each antibody staining and their controls in the manuscript.
Moreover, they also have to report the amount of slide of each tissue analyzed, their location in the organ and the amount of animals used, in order to know whether the quantification performed is accurate. The number of cells count for each staining should be higher than the number obtained by the formula (standard deviation•0.83/mean•0.05)2.
Authors have to report the statistical test performed.

Reviewer 4 Report

The manuscript is good and theoretically sound, but still in need of some English-language editing. Results and Figures are well constructed, and photomicrographs are illustrative.

The authors showed that the 2 ERs and PRs and pituitary play a role in the reproductive cycle of T. compressicauda.

Some recommended changes:

L52 The genital tract...

L56 ...during sexual quiescence...

L 65 ...mucus... (used as a noun here; "mucous" is the adjective)

L128 ...bovine serum albumin...

LHR, FSHR, and PRLR detection methods are appropriate.